# Placebos without deception reduce self-report and neural measures of emotional distress

Darwin A. Guevarra [1,2✉], Jason S. Moser[2], Tor D. Wager [3,4] & Ethan Kross [1]

Several recent studies suggest that placebos administered without deception (i.e., non-deceptive placebos) can help people manage a variety of highly distressing clinical disorders and nonclinical impairments. However, whether non-deceptive placebos represent genuine psychobiological effects is unknown. Here we address this issue by demonstrating across two experiments that during a highly arousing negative picture viewing task, non-deceptive placebos reduce both a self-report and neural measure of emotional distress, the late positive potential. These results show that non-deceptive placebo effects are not merely a product of response bias. Additionally, they provide insight into the neural time course of non-deceptive placebo effects on emotional distress and the psychological mechanisms that explain how they function.

[1] Department of Psychology, University of Michigan, Ann Arbor, 530 Church St., Ann Arbor, MI 48109, USA. [2] Department of Psychology, Michigan State University, 316 Physics Rd., East Lansing, MI 48824, USA. [3] Department of Psychology and Neuroscience, University of Colorado, Boulder, 1905 Colorado Ave., Boulder, CO 80309, USA. [4] Department of Psychology and Brain Sciences, Dartmouth College, 3 Maynard St., Hanover, NH 03755, USA. ✉email: guevarra@umich.edu

Placebo interventions offer a cost-effective tool to manage a host of clinical disorders and nonclinical symptoms[1–3]. However, an important ethical issue prevents their widespread use: the ubiquitous belief that for placebos to be effective, a person needs to be deceived into believing they are taking an active treatment[4,5]. Recently, researchers have begun to examine whether the beneficial effects of placebos can be harnessed without deception by communicating to participants what placebos are, explaining the science behind how they work, and highlighting how placebos can still provide beneficial effects even if people know they are taking them[4,5]. This verbal suggestion approach leverages one of the primary psychological mechanisms through which placebos operate: a person's expectation that their condition will improve after receiving a treatment[6–8].

Guided by this approach, researchers have demonstrated the beneficial effects of non-deceptive placebos for a variety of conditions, including irritable bowel syndrome[5], chronic back pain[9], experimental pain[10,11], and emotional distress, psychological well-being, and sleep quality[12] (see Supplementary Note 1 for the distinction between open-label and non-deceptive placebos). However, these studies have primarily documented the benefits of non-deceptive placebos using self-report measures[13–16]. Out of twenty-six published non-deceptive placebo studies to date, eight included objective behavioral or biological measures. Only one of these eight studies showed an effect on behavioral outcomes, and no direct effects on biological outcomes have been documented[10,17–21] (see Supplementary Table 1 for a current list of non-deceptive placebo studies). Therefore, it remains unclear whether the beneficial effects associated with non-deceptive placebos represent genuine psychobiological effects[2,3].

Here, we argue that prior research may have failed to observe non-deceptive placebo effects on objective biological measures because they focused on domains (e.g., wound healing recovery rate or physical skin reaction) that do not reliably respond to deceptive placebos induced through verbal suggestion[22–25]. Put simply, if a deceptive placebo induced through verbal suggestion does not reliably impact biological outcomes in these contexts, there is no reason to expect a non-deceptive placebo should either.

Guided by this logic, we examine whether non-deceptive placebos can reduce self-report measures and objective biological markers in a context that is responsive to deceptive placebo effects: emotional distress[26–33]. In Experiment 1 ($n = 68$), we examine the effect of a non-deceptive placebo manipulation on self-report emotional distress in response to viewing negative emotional images (see Fig. 1a for task sequence). In Experiment 2 ($n = 218$), using a similar image viewing paradigm (see Fig. 2a for task sequence), we examine the effect of the same non-deceptive placebo manipulation on a neural biomarker of emotional distress: the late positive potential (LPP).

The LPP is an electroencephalogram (EEG) derived event-related brain potential (ERP) response that measures millisecond changes in the neural activity involved in emotional processing[34]. The early-time window of the LPP (400–1000 ms) indexes attention allocation[34]; the sustained time window (1000–6000 ms) indexes conscious appraisals and meaning-making mechanisms involved in emotion processing[34,35] and is consistently downregulated by cognitive emotion regulation strategies[36–39]. Consistent with its role in immediate attentional orienting responses to emotional stimuli and later appraisal processes, neural sources of the LPP include both the amygdala[35,40] and dorsolateral prefrontal cortex[41]. Thus, the LPP is ideally suited to help examine the neural mechanisms and time course of non-deceptive placebo effects on emotional distress.

In both experiments, we randomly assigned participants to either a non-deceptive placebo group or a control group.

Participants in the non-deceptive placebo group read about placebo effects and were then asked to inhale a nasal spray consisting of saline solution. They were told that the nasal spray was a placebo that contained no active ingredients, but would help reduce their negative emotional reactions to viewing distressing images if they believed it would. Participants in the control group read about the neural processes underlying the experience of pain and were also asked to inhale the same saline solution spray; however, they were told that the purpose of the nasal spray was to improve the clarity of the physiological readings we were recording in the study. The articles were matched for narrative structure, emotional content, and length (see "Methods" section, Supplementary Methods 1 and 2 for details).

Consistent with the idea that non-deceptive placebos reflect genuine psychobiological effects, we hypothesized that the non-deceptive placebo group (vs. control) would report less negative affect and exhibit lower neural activity during the sustained LPP time window. Given conflicting evidence concerning how deceptive placebos influence attentional processes, we were agnostic about how non-deceptive placebos would influence early LPP amplitude.

In Experiment 1, we find that non-deceptive placebos (vs. control) reduces self-report measures of emotional distress. Moreover, in Experiment 2, we demonstrate that non-deceptive placebos (vs. control) also reduces neural activity during the sustained LPP time window that indexes meaning-making stages of emotional reactivity. We do not find any effects of non-deceptive placebos (vs. control) on attentional processes as indexed by the early LPP time window. In summary, non-deceptive placebos can downregulate both self-report and neural measures of emotional distress, providing evidence that they are more than response bias.

## Results

**Self-report emotional distress in Experiment 1**. A 2 (condition: control and non-deceptive placebo) × 2 (picture type: neutral and negative) mixed-factorial ANOVA revealed significant main effects of condition, $F(1, 60) = 7.34$, $p = 0.009$, $\eta_\rho^2 = 0.109$, and picture type, $F(1, 60) = 627.25$, $p < 0.001$, $\eta_\rho^2 = 0.913$, indicating that participants in the non-deceptive placebo group (vs. control) reported less emotional distress, and viewing negative (vs. neutral pictures) generated more distress.

These main effects were qualified by a significant condition by picture type interaction, $F(1, 60) = 12.41$, $p < 0.001$, $\eta_\rho^2 = 0.171$. As Fig. 1b illustrates, participants in the non-deceptive placebo group reported less distress after viewing negative pictures compared to participants in the control group, $t(60) = 3.94$, $p = 0.0002$, $d = 1.00$. There was no non-deceptive placebo effect on neutral pictures, $t(60) = -0.36$, $p = 0.72$, $d = -0.09$. Supplementary Table 2 reports exploratory correlational analysis regarding beliefs, expectations, and self-reported emotional distress. These findings demonstrate that the non-deceptive placebo manipulation we administered is effective at reducing subjective emotional distress. Experiment 2 examined whether this emotion-dampening effect generalizes to an objective neural biomarker of emotional reactivity.

**Sustained LPP in Experiment 2**. We tested our predictions concerning whether non-deceptive placebos would influence an objective neural biomarker of emotional distress by performing a mixed-factorial ANOVA on the sustained LPP using a broad set of topographically organized clusters of electrodes that have been the focused of prior work[38,42,43] (see "Methods" section for details of our preregistered data analytic approach). As expected, the non-deceptive placebo manipulation led to a significant

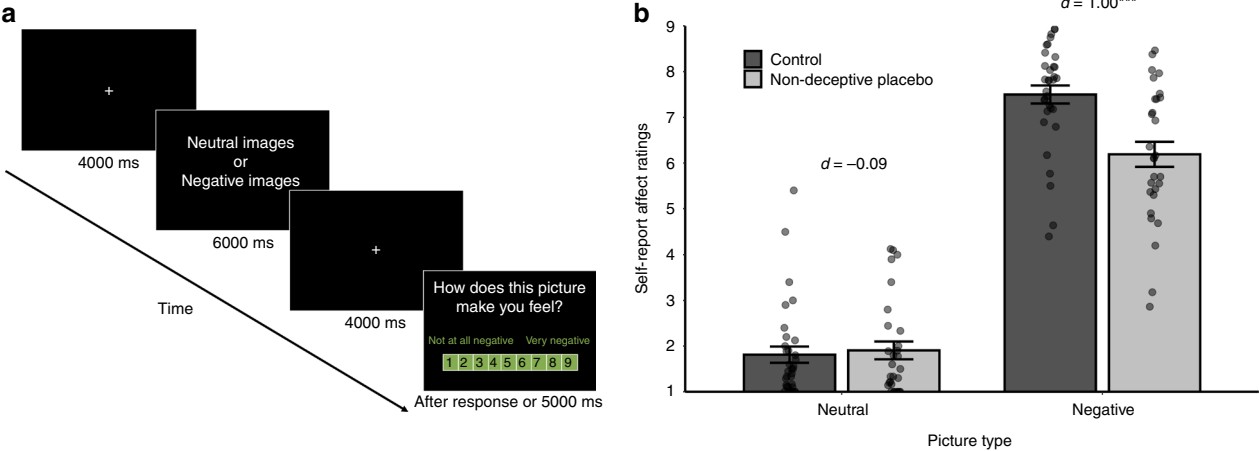

**Fig. 1 Experiment 1 trial sequence and results ($n = 62$ participants). a** All participants viewed a total of 40 images (30 negative and 10 neutral) in one block, and reported how they felt after each image on a nine-point scale from 1 (not at all negative) to 9 (very negative). **b** A mixed-factorial ANOVA (condition by picture type) was conducted followed-up by independent pairwise comparisons for relevant contrasts. All tests were two-tailed, and follow-up tests were not adjusted for multiple comparisons. Bars represent the mean self-report ratings calculated for condition (control group, $n = 33$, and non-deceptive placebo group, $n = 29$) per picture type (neutral and negative). Error bars represent ± 1 SEM. Dots represent mean values for each participant per picture type. There was a significant interaction between condition and picture type ($p = 0.0008$). Follow-up tests showed no difference in emotional distress ratings between the control group and non-deceptive placebo group for neutral pictures ($p = 0.72$); however, the non-deceptive placebo group, compared to the control group, reported less emotional distress when viewing negative pictures ($p = 0.0002$). No asterisk = not significant, ***$p < 0.001$.

reduction in a neural biomarker of emotional distress, as evidenced by a main effect of condition on sustained LPP amplitude, $F(1, 194) = 8.98$, $p = 0.003$, $\eta_\rho^2 = 0.044$.

This main effect was qualified by a significant condition by time interaction, $F(1.62, 314.94) = 4.58$, $p = 0.017$, $\eta_\rho^2 = 0.023$. As Fig. 2b illustrates, participants in the non-deceptive placebo group showed a gradual reduction in sustained LPP amplitude throughout the picture presentation, as shown by a significant time effect in the non-deceptive placebo group, $F(1.73, 167.98) = 6.38$, $p = 0.003$, $\eta_\rho^2 = 0.062$, and followed by a within-subjects linear contrast, $F(1, 97) = 6.83$, $p = 0.01$, $\eta_\rho^2 = 0.066$. In comparison, the sustained LPP amplitude for participants in the control group did not change in magnitude throughout the picture presentation, $F(1.53, 148.63) = 0.41$, $p = 0.61$, $\eta_\rho^2 = 0.004$. Figure 2c shows topographic headmaps of the sustained LPP activity across the scalp (with amplitude from neutral and negative images collapsed) broken down by condition and time. Figure 2d illustrates that the magnitude of the difference between the control group and the non-deceptive placebo group increased at ~2000–3000 ms, then peaked and plateaued at ~3000–4000 ms. See Supplementary Table 3 for detailed independent pairwise comparison statistics.

To corroborate these findings, we performed a similar analysis at CPz, where the sustained LPP is typically maximal. We found similar patterns for the main effect of condition and condition by time interaction (see Supplementary Fig. 1, Supplementary Table 4). We also report other significant interactions with condition in Supplementary Fig. 2, any condition by sample interactions in Supplementary Note 2, and exploratory correlational analysis regarding beliefs, expectations, and sustained LPP activity in Supplementary Table 5.

There was no significant three-way interaction among condition, picture type, and time, $F(1.88, 364.35) = 0.08$, $p = 0.91$, $\eta_\rho^2 < 0.001$, nor were there any other significant interactions involving condition and picture type ($p > 0.05$). These null interactions involving condition and picture type suggest that the non-deceptive placebo manipulation exerted a general dampening effect on emotional reactivity in response to both neutral and

negative images. As we elaborate in more detail in the "Discussion" section, although this pattern is inconsistent with the self-report findings we observed in Experiment 1, it is consistent with several placebo studies that have shown a main effect of deceptive placebos across neutral and negative stimuli for autonomic and neural measures[27,28].

**Early LPP in Experiment 2.** The early LPP (400–1000 ms) indexes attention allocation to incoming emotional stimuli[33,44]. As noted earlier, we did not have strong predictions regarding how the non-deceptive placebo we administered would affect the early LPP because prior research provides mixed evidence regarding how deceptive placebos influence attention allocation processes. While some studies suggest that deceptive placebos amplify attention to negative stimuli, others suggest the opposite[27,33,45,46].

We examined the effects of non-deceptive placebos on the attentional stages of emotional processing by performing a mixed-factorial ANOVA on the same broad set of topographically organized clusters of electrodes, but focused on the early LPP time window (400–1000 ms; see "Methods" section for details of our preregistered data analytic approach)[34]. This analysis revealed a complicated set of interactions, such as a three-way condition by time by anterior/posterior interaction $F(1, 194) = 4.00$, $p = 0.047$, $\eta_\rho^2 = 0.02$, as well as a five-way condition by picture type by time by hemisphere by anterior/posterior interaction $F(1, 194) = 4.46$, $p = 0.036$, $\eta_\rho^2 = 0.022$. Probing these interactions did not, however, reveal any consistent condition effects (see Supplementary Table 6 and Supplementary Table 7 for contrasts). Moreover, we did not detect any condition effect at CPz, where the early LPP is typically maximal (all $p > 0.05$). We did not further probe any significant interactions with sample and condition for the early LPP. In summary, we found no reliable non-deceptive placebo effect on the early LPP.

## Discussion
The beneficial effects of non-deceptive placebos have been established in self-report measures for a host of clinical conditions and nonclinical impairments[4,47]. However, it is unclear

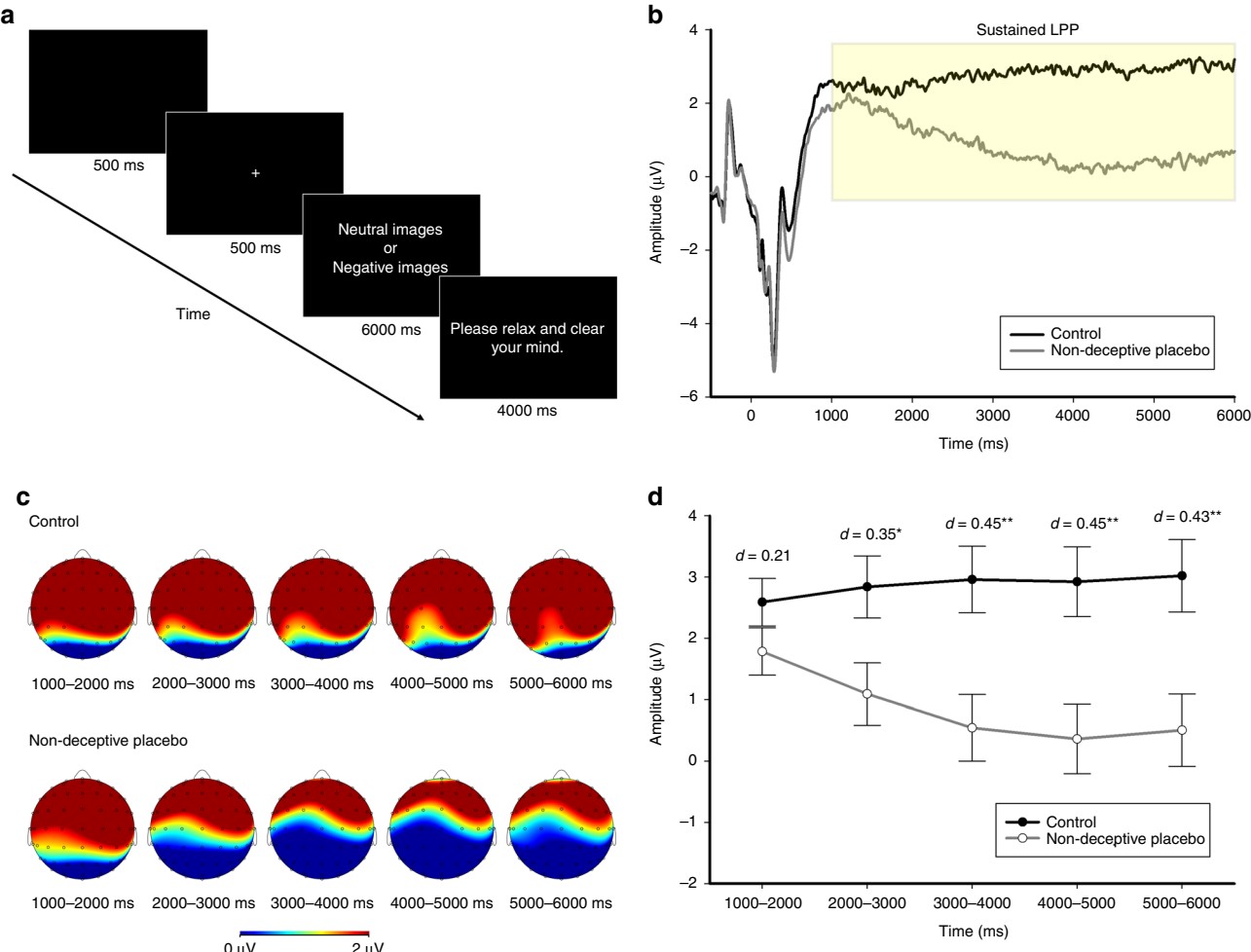

**Fig. 2 Experiment 2 trial sequence and results ($n = 198$ participants). a** Combining two samples, all participants viewed a total of 60 pictures (30 negative and 30 neutral) broken down in two blocks, while their continuous EEG, measured in microvolts (μV), was recorded. **b** Picture-locked ERP waveforms depict estimated mean amplitude for the control group ($n = 99$) and non-deceptive placebo group ($n = 99$), with amplitudes from neutral and negative images across sites collapsed for each group. A higher number indicates a larger amplitude. Picture onset occurs at 0 ms. The yellow shaded area represents the time window for analysis of the sustained LPP. **c** Headmaps depict the topographical distribution of the sustained LPP with activity from neutral and negative images collapsed, and separated by condition and time. Red indicates a larger LPP amplitude, and blue indicates a smaller LPP amplitude. **d** Combining two samples, a mixed-factorial ANOVA (condition by sample by picture type by time by hemisphere by anterior/posterior by inferior/superior) was conducted followed-up by independent pairwise comparisons for relevant interactions. All tests were two-tailed, and follow-up tests were not adjusted for multiple comparisons. There was a significant interaction between condition and time ($p = 0.017$). Line plots depict estimated marginal mean amplitude for the control group ($n = 99$) and non-deceptive placebo group ($n = 99$), with activity from picture type collapsed across time (1000–2000 ms, 2000–3000 ms, 3000–4000 ms, 4000–5000 ms, and 5000–6000 ms). Error bars represent ± 1 SEM. Follow-up tests showed that the non-deceptive placebo group exhibited a gradual decrease in sustained LPP amplitude compared to the control group across the five time points: 1000–2000 ms ($p = 0.546$), 2000–3000 ms ($p = 0.016$), 3000–4000 ms ($p = 0.002$), 4000–5000 ms ($p = 0.002$), and 5000–6000 ms ($p = 0.003$). Positive $d$ indicates beneficial effects of non-deceptive placebos. See Supplementary Table 3 for detailed comparison statistics. No asterisk = not significant, *$p < 0.05$, **$p < 0.01$.

whether they can affect objective biomarkers, such as neural responses that are relevant to emotion generation and emotional disorders. Our studies help address this issue and provide important insights into the time course of the neural mechanisms underlying non-deceptive placebo effects on emotional distress.

Consistent with the deceptive placebo literature, we show that non-deceptive placebos decreased self-report measures of emotional distress[26,27,29–33,46]. More importantly, we demonstrate that non-deceptive placebos decreased an objective neural marker of emotional distress during the appraisal stages of emotional processing: the sustained LPP. This finding provides initial support that non-deceptive placebos, at least in the domain of

emotional distress, are not merely a product of response bias, but represent genuine psychobiological effects.

These findings also help illuminate the neural time course of non-deceptive placebo effects on emotional distress. It seems that non-deceptive placebos do not exert their regulatory effects immediately and require some time to reduce emotional reactivity (Fig. 2). This pattern of gradual decreases in sustained LPP amplitude throughout the picture presentation is consistent with the time course of deceptive placebo effects on pain processing, where participants in the placebo condition initially experience equivalent levels of pain similar to the control condition before it is modulated by the placebo intervention[48]. A gradual decrease in sustained LPP amplitude appears to happen at ~2000–3000 ms and then plateaus at the

3000–4000 ms range, following a non-deceptive placebo intervention. This time course pattern in neural activity suggests that non-deceptive placebos are likely acting on appraisal and meaning-making mechanisms[49,50]. Moreover, it raises questions about what type of appraisal processes are occurring when someone receives a non-deceptive placebo intervention and the degree to which these appraisals are conscious or unconscious processes.

Consistent with prior research[27,30], we observe an asymmetry in terms of how our non-deceptive placebo manipulation impacted participants' self-report and EEG measures of emotional distress. In Experiment 1, we show a non-deceptive placebo effect for negative stimuli but not neutral stimuli; however, in Experiment 2, we observe a non-deceptive placebo effect for both neutral and negative stimuli. One explanation for this asymmetry may have to do with the temporal feature of self-report and the sustained LPP. The sustained LPP measures online reactions to the images while self-report is assessed retrospectively, 4000 ms after picture offset. It may be that by the time participants are asked about the neutral images, any small negative emotional distress they experienced may have returned to baseline levels. More broadly, these findings are compatible with a large body of research suggesting that self-report, behavior, peripheral physiological, and neural measures are not redundant and often do not cohere[51,52]. Taken together, these findings underscore the importance of examining the effects of non-deceptive placebos across multiple levels of analysis.

Our findings also have important translational implications. Acute episodes of emotional distress have relevance not only for daily emotional life, but for many physical and psychiatric conditions[1,53,54]. In terms of physical conditions, emotional distress is associated with increased chances of chronic pain onset and amplifying existing pain experience[55]. As such, non-deceptive placebos can help manage the emotional aspect of many medical conditions that have a pain component. For psychiatric conditions, non-deceptive placebos may be used as cointerventions with existing therapies, especially for disorders in which emotion dysregulation is a core feature, such as depression and anxiety[54]. From a nonclinical standpoint, non-deceptive placebos also offer an alternative emotion regulation strategy that some researchers believe are distinct from internally generated reappraisal strategies, which often require intact cognitive control mechanisms and available cognitive resources[50,56–58]. We believe these various clinical and nonclinical areas provide important translational research directions.

It is important to acknowledge that we did not find a significant relationship between beliefs and expectations with self-report (Supplementary Table 2), and neural measures of emotional distress (Supplementary Table 5). This lack of a relationship seems to be consistent with the broader non-deceptive placebo literature, since associations between expectations and outcome measures are not commonly documented and observed. In fact, of the twenty-six non-deceptive placebo studies published thus far, eleven report asking expectation questions, and only two show a relationship[12,19]. These inconsistencies may be due to the well-established finding that people frequently lack direct access to internal states, making it difficult to provide an accurate account of their expectations[59]. Future theoretical and empirical work is needed to delineate the factors that influence the associations between expectations and outcome measures.

Future research is also needed to examine how these findings generalize to other demographics. We sampled from a population of college students, limited in age range and not ethnically diverse. Moreover, because of sex differences between males and females in emotional reactivity, Experiment 2 only recruited female participants to minimize the confounding effect of sex[60]. An important question for future research is to examine if the sex of the participant influences the efficacy of non-deceptive placebos on emotional distress and other domains.

Non-deceptive placebos may offer a cost-effective intervention to help manage a host of clinical disorders and nonclinical symptoms[4,61]; however, it is important first to establish that their beneficial effects go beyond self-report measures and lead to positive changes on objective biological markers[47]. Our findings demonstrate an objective non-deceptive placebo effect on a neural biomarker that is relevant for emotion regulation and conditions characterized by emotional distress. Future research should examine the generalizability of these findings to other populations, domains, and biomarkers.

## Methods

**Participants**. For Experiment 1, participants were recruited from a nonclinical sample at a large university in the Midwest. They were compensated with course credit. Sixty-eight participants participated in the study, but six were removed due to experimenter error or substantial deviation from the protocol ($n = 3$), participant indicating they were a non-native English speaker at the exit survey ($n = 1$), participant indicating that they misread the self-report scale ($n = 1$), and software error resulting in no self-report affective ratings ($n = 1$). Four were removed from the non-deceptive placebo group, and two were removed from the control group. The final sample submitted to analyses included 62 participants with $n = 33$ in the control group ($M_{age} = 18.61$, SD = 0.83; 39.4% female; 60.6% European American) and $n = 29$ in the non-deceptive placebo group ($M_{age} = 18.76$, SD = 0.74; 34.5% female; 75.9% European American). Experiment 1 complied with all relevant ethical guidelines and regulations involving human participants, and was approved by the University of Michigan's Institutional Review Board. All participants provided informed consent before participating.

For Experiment 2, participants were recruited from a nonclinical sample at another large university in the Midwest. Two samples were collected, sample 1 ($n = 115$) and sample 2 ($n = 103$). They were compensated with course credit ($n = 110$) or $20 ($n = 108$) for their time. All participants were female to control for sex differences in brain structure, brain formation, emotion processing, and emotion regulation ability[52,62,63]; moreover, all participants were right-handed, between the ages of 18 and 30, and native English speakers. Participants who reported a history of severe mental illness or seizures were excluded. Participants recruited through a course credit system and did not meet all of our criteria were automatically filtered out, and were not able to sign up for the study. Participants recruited through a payment system were sent a screening survey, and eligible participants were scheduled to come into the lab.

A total of 218 people participated in Experiment 2. Twenty participants were removed from analysis due to reporting that English was not their native language at the exit survey ($n = 1$), software error ($n = 4$), and excessive artifacts due to eye and body movements ($n = 15$). One hundred and ninety-eight participants were submitted to analyses, with $n = 99$ in the control group ($M_{age} = 19.92$, SD = 2.14; 78.8% European American) and $n = 99$ in the non-deceptive placebo group ($M_{age} = 19.78$, SD = 2.36; 80.8% European American). Experiment 2 complied with all relevant ethical guidelines and regulations involving human participants, and was approved by Michigan State University's Institutional Review Board. All participants provided informed consent before participating.

**Experimental design**. For both experiments, participants were told that the study was on cognitive processing, memory, and emotion. Participants were randomly assigned to a control or non-deceptive placebo group (see Supplementary Fig. 3 for a design diagram). Those in the control group read an article on the neurological processes of pain and how to treat it (Supplementary Methods 1). Those in the non-deceptive placebo group read an article on the placebo effect, how powerful it is for some conditions, and how it can still work even without deception (Supplementary Methods 1). After reading the articles, the experimenter delivered different nasal spray instructions to the control and non-deceptive placebo participants. For the non-deceptive placebo group, the experimenter summarized the main points of the reading, positively framed that placebos can still work if the participant believes it will, and administered a saline nasal spray once to each nostril. For the control group, the experimenter explained that the saline nasal spray was designed to help obtain better physiological readings (Supplementary Methods 2). The articles were matched for narrative structure, negatively valenced words (control = 62, non-deceptive placebo = 58), and length (control = 1287 words, non-deceptive placebo = 1270 words; see Supplementary Methods 1 and 2 for details). Participants in the control and non-deceptive placebo group did not differ in terms of reading duration, writing duration, perception of article quality (all $p > 0.05$), and post article reading mood (Experiment 1, $p > 0.05$; see Supplementary Tables 8 and 9 for details).

For the non-deceptive placebo group, the experimenter and the participant were not blind to the condition since our manipulation involved honestly telling participants they were receiving a placebo. Nevertheless, it is important to highlight that those in the non-deceptive placebo group were not aware they were receiving a placebo nasal spray until the actual nasal spray administration. This feature of our

design reduces the bias from participants knowing they are participating in a study involving placebos before coming into the lab. Equally important, unlike previous work on non-deceptive placebos, the control group was blind to their condition and was not aware they were participating in a placebo study or that they were in the control group[5]. This feature of our experiments reduces the bias that stems from a participant knowing they are in the control group and will not receive the experimental treatment[64].

**Image viewing task**. After the nasal spray administration, participants engaged in an image viewing task. For Experiment 1, participants viewed one block of forty images (30 negative and 10 neutral; see Supplementary Table 10 for a complete list of these images) based on their normative valence and arousal ratings. The block design was based on a previous placebo study on emotional distress[29]. The negative images were considered high intensity with a $M_{valence} = 2.30$ (1 = very unpleasant; 9 = highly pleasant) and $M_{arousal} = 6.37$ (1 = low; 9 = high)[65,66]. The images were presented in a randomized order in forty trials using E-Prime (version 2.0; Psychology Software Tools, Pittsburg, PA). For each image, participants viewed a fixation cross (4000 ms), a random image (6000 ms), and another fixation cross (4000 ms), followed by an affective rating period (5000 ms or less depending on when the participant chose their response). For each image, the participant rated how the picture made them feel on a nine-point Likert scale from 1 (not at all negative) to 9 (very negative; see Fig. 1a for trial sequence). A mean score for neutral and negative images was calculated from individual ratings. See Supplementary Note 3 for an explanation regarding skin conductance response data.

For Experiment 2, participants viewed a total of sixty images (30 negative images, $M_{valence} = 2.30$, $M_{arousal} = 6.37$; 30 neutral images, $M_{valence} = 4.92$, $M_{arousal} = 2.85$), divided into two blocks of 30 images (15 negative and 15 neutral) based on previous work on emotion regulation and the LPP[38,65] (see Supplementary Table 11 for a complete list of these images). The nasal spray was administered twice to each nostril before each block. The mean valence and arousal ratings for each block were matched, and did not significantly differ from each other ($p > 0.05$). The pictures were presented in a randomized order using E-Prime (version 2.0; Psychology Software Tools, Pittsburgh, PA). For each image, participants viewed a blank screen (500 ms), a fixation cross (500 ms), a random image (6000 ms), and a relaxation prompt instructing them to relax and clear their mind (4000 ms; see Fig. 2a for trial sequence). Critically, participants did not self-report their negative feelings after each trial or after each block to obtain pure neural signals of emotional reactivity without intervening introspective questions[37,42].

**Data analytic strategy for Experiment 1**. All statistical analyses for Experiment 1 were performed with SPSS (version 26), and Fig. 1b bar graph was created with R studio (version 3.6.1) and ggplot2 (version 3.3.0)[67]. For the primary analysis, we performed a mixed-factorial ANOVA with condition (control and non-deceptive placebo) as a between-subjects factor, and picture type (neutral and negative) as a within-subjects factor. A significant interaction between condition and picture type was followed by independent pairwise comparisons contrasting control minus non-deceptive placebo for neutral and negative pictures. Follow-up comparisons did not use any adjustments for multiple comparisons. For preliminary analyses, separate independent-samples $t$-tests were conducted for each respective variable (Supplementary Table 8). All tests were two-tailed and used a significance level of $p < 0.05$. Partial eta squared was calculated for all ANOVA results, and Cohen's $d$ was calculated for all $t$-tests.

**Psychophysiological recording and data reduction for Experiment 2**. Continuous EEG activity was recorded using the ActiveTwo Biosemi system (Biosemi, Amsterdam, the Netherlands) from a 64-electrode cap arranged according to the International 10–20 system. Two additional electrodes were placed on the left and right mastoids for use in offline references. Three additional electrodes were placed inferior to the left pupil, and the left and right outer canthi were used to record blinks and eye movements. A common mode sense active electrode and driven right leg passive electrode formed a ground specified by the Biosemi system; this limited the amount of current that could return to the participant. Bioelectric signals were sampled at 1024 Hz.

EEG signal processing and creating topographic headmaps for Fig. 2c were performed using BrainVision Analyzer (version 2.2; BrainProducts, Gilching, Germany). Each electrode recording was referenced to the mean of the mastoids, band-pass filtered (cutoffs: 0.01–20 Hz; 24 dB/oct roll-off), and subjected to ocular artifact correction[68]. Each picture trial was subjected to standard artifact rejection procedures using a computer-based algorithm criterion: a voltage step exceeding 50 μV between contiguous sampling points, a voltage difference of 400 μV within a trial, and a maximum voltage difference of <0.5 μV within 100 ms intervals. The average activity 500 ms before picture onset served as a baseline and was subtracted from each data point after picture onset.

**Data analytic strategy for Experiment 2**. The LPP is characterized by a broad and sustained waveform that has an early-time window (400–1000 ms), which indexes attention allocation, and a later sustained time window (1000–6000 ms), which indexes appraisal and meaning-making stages[34,38,42,69,70]. The LPP is larger for highly arousing negative and positive stimuli compared to neutral ones[34,37]. To analyze the LPP, we reduced the data into topographically organized clusters of electrodes based on prior work[42,43]. We computed a mean for eight clusters using the average of each specific electrode: Left-Anterior-Superior (AF3, F1, F3, FC1, and FC3), Right-Anterior-Superior (AF4, F2, F4, FC2, and FC4), Left-Anterior-Inferior (AF7, F5, F7, FC5, and FT7), Right-Anterior-Inferior (AF8, F6, F8, FC6, and FT8), Left-Posterior-Superior (CP1, CP3, P1, P3, and PO3), Right-Posterior-Superior (CP2, CP4, P2, P4, and PO4), Left-Posterior-Inferior (CP5, P5, P7, PO7, and TP7), and Right-Posterior-Inferior (CP6, P6, P8, PO8, and TP8). For the sustained LPP, the time window was broken down into 1000 ms epochs. For the early LPP, the time window was broken down into 300 ms epochs.

We elected to preregister our data analytic plan on AsPredicted.org (http://aspredicted.org/blind.php?x=ie6r5j). All analyses for Experiment 2 were performed with SPSS (version 26), and Fig. 2b, d, Supplementary Fig. 1, and Supplementary Fig. 2 were created with SigmaPlot (version 14). Combining two samples, we first examined the effects of non-deceptive placebos on the sustained LPP by performing a 2 (condition: control and non-deceptive placebo) × 2 (sample: sample 1 and sample 2) × 2 (picture type: neutral and negative) × 5 (time: 1000–2000 ms, 2000–3000 ms, 3000–4000 ms, 4000–5000 ms, and 5000–6000 ms) × 2 (hemisphere: left and right) × 2 (anterior/posterior: anterior and posterior) × 2 (inferior/superior: inferior and superior) mixed-factorial ANOVA with condition and sample as a between-subjects factor, and the other variables as a within-subjects factor. We focused on the main effect of condition and any interaction effects involving condition that were robust against sample type. Greenhouse–Geiser corrections were applied to relevant interaction analyses. We report outlier detection procedures and additional robust analysis found in Supplementary Table 12. Moreover, to corroborate this analysis, we performed a 2 (condition: control and non-deceptive placebo) by 2 (sample: sample 1 and sample 2) × 2 (picture type: neutral and negative) × 5 (time: 1000–2000 ms, 2000–3000 ms, 3000–4000 ms, 4000–5000 ms, and 5000–6000 ms) at CPz, where the LPP is typically maximal and analyzed. We report the analysis at CPz in Supplementary Fig. 1. We also report any significant interactions with condition in Supplementary Fig. 2 and any condition by sample interaction in Supplementary Note 2.

Next, we tested the effect of non-deceptive placebos on the early LPP (400–1000 ms) by performing a 2 (condition: control and non-deceptive placebo) × by 2 (sample: sample 1 and sample 2) × 2 (picture type: neutral and negative) × 2 (time: 400–700 ms and 700–1000 ms) × 2 (laterality: left and right) × 2 (anterior/posterior: anterior and posterior) × 2 (inferior/superior: inferior and superior) mixed-factorial ANOVA with condition and sample as between-subjects factors, and the other variables as within-subjects factors. We focused on the main effect of condition and any interaction effects involving condition that were robust against sample type. To corroborate this analysis, we perform a 2 (condition: control and non-deceptive placebo) by 2 (sample: sample 1 and sample 2) × 2 (picture type: neutral and negative) × 2 (time: 400–700 ms and 700–1000 ms) at CPz, where the LPP is typically maximal and analyzed.

Any significant interactions with condition were probed further until it could be followed by independent pairwise comparisons. Follow-up comparisons did not use any adjustments unless otherwise stated. For preliminary analyses, separate independent-samples $t$-tests were conducted for each respective variable (Supplementary Table 9). All tests were two-tailed and used a significance level of $p < 0.05$. Partial eta squared was calculated for all ANOVA tests, and Cohen's $d$ was calculated for all $t$-tests.

**Questionnaires**. For both experiments, participants completed additional measures, such as duration of reading and writing time, quality of the article readings, and belief in the effectiveness of placebos without deception (See Supplementary Methods 3 and 4). Participants in Experiment 2 completed extra measures, such as the perception of the experimenters and individual difference measures, such as the tendency to worry, trait anxiety, levels of optimism, and proneness to social desirability responding (See Supplementary Methods 4). Preliminary analyses and results for Experiment 1 are reported in Supplementary Table 8, and those for Experiment 2 are reported in Supplementary Table 9.

**Reporting summary**. Further information on research design is available in the Nature Research Reporting Summary linked to this article.

## Data availability

Data supporting these findings can be found at the Open Science Framework (https://osf.io/s3b8d/). SPSS (version 26) is used for all statistical analyses. Data and R Code underlying Fig. 1b can be found in Experiment 1 data files. Data and SPSS syntax underlying Fig. 2d, Supplementary Fig. 1b, and Supplementary Figs. 2a, b can be found in Experiment 2 data files. A reporting summary for this article is available as a Supplementary Information file. Additional data from these studies are available from the corresponding author upon request.

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

## Acknowledgements
We thank members of the Emotion and Self-Control Laboratory for helping set up Experiment 1 and collect data. We thank members of the Clinical Psychophysiology Laboratory, especially Megan Fisher, Courtney Callahan, Connor Dillivan, and William Eckerle, for helping set up Experiment 2 and collect data. We thank Kerby Shedden and Nicholas Michalak for their statistical guidance.

## Author contributions
D.A.G. designed the experiment, supervised data collection, analyzed the data, interpreted the results, and wrote and revised the paper. J.S.M. and E.K. designed the experiment, supervised data collection, interpreted the results, and wrote and revised the paper. T.D.W. interpreted the results and wrote and revised the paper.

## Competing interests
The authors declare no competing interests.
