## [Peer Review File · Nature Communications]

Reviewers' Comments:

Reviewer #1:

Remarks to the Author:

This is an elegant study that addresses an important topic in affective neuroscience: are non-deceptive placebo effects on negative emotional responses "true" effects or do they simply reflect the demand characteristics of the task (i.e., if subjects say they feel less pain or distress from a non-deceptive placebo manipulation, do they simply say what they believe the experimenter wants to hear)? The key to answering this question is to provide evidence that the non-deceptive placebo condition affects objective behavioral or biological correlates of the to-be-modulated emotional response.

The authors have conducted a thorough and well-designed investigation of the topic that involves two studies where a non-deceptive placebo group is compared to a control group not receiving any explicit expectation manipulation in their responses to negative and neutral affective pictures. Study 1 provides only self-report ratings of negative affect, study 2 provides only EEG measures (Late Positive Potential, LPP). The paper is well written, the argumentation is clear and convincing, the statistical analyses are suitable and rigorous. The results demonstrate an effect of the placebo manipulation on the EEG.

I nevertheless have two major concerns that I feel the authors need to address before I can make an informed recommendation.

1. Specificity

a) Subjects in the control condition are made to read an article about pain before the experiment. I was wondering if this might induce a general negative emotional state or even negative expectations about the effects of negative or unpleasant emotional material on well-being, distress, or health. In other words: could the control condition effectively be a distress or even nocebo condition? (In which case the differences in ratings and EEG measures between the two groups could be driven by amplifying effects on affective responses in the control group rather than by dampening effects in the placebo condition.)

b) In relation to this: I would be more confident that the group differences in rating and EEG are really due to the positive expectations raised by the placebo manipulation if the authors could demonstrate a correlation of rating/EEG effects with individual or group differences in the ratings of subjects' beliefs in the effects of non-deceptive placebos as well as in the effects of the nasal spray that the subjects provided at the end of the instruction phase (see Supplement p. 13 ff.). Supplementary Tables 2 and 3 show that these mostly differ between groups and also exhibit substantial variability within groups, and the numbers of subjects included in the two studies (62 and 100, resp.) are clearly sufficient for performing individual-differences analyses. Hence, these analyses are feasible. Showing a relationship between expectations and placebo effects is also more or less standard in placebo studies.

c) Another related point: Supplementary Table 3 also shows that subjects in the control group of study 2 perceived experimenters as warmer and more competent. This is a surely unexpected effect of the manipulation that again may have had an impact on EEG in this

case. The authors themselves think that this finding suggests "that there may be something about our non-deceptive placebo manipulation (i.e., verbal suggestion instructions during the nasal spray instructions) that decreases perception of warmth and competence. This may have influenced the effectiveness of our manipulation ...". Hence, there is a clear need to disambiguate better the potential sources of the described differential EEG effect, which could be done by some kind of multiple regression including all the different ratings. [Note that the authors believe the reduced perceptions of experimenters' warmth and competence in the placebo group may have "reduced the effects we observed" (Suppl. Table 3). They thereby seem to suggest that this could only have made their EEG-based assay of the placebo effect less sensitive, but not have produced false positive findings. However, their EEG effects go both ways – relatively reduced responding in the placebo group in the sustained LPP (Fig. 2) and relatively enhanced responding in the early LPP (Fig. 3); hence, the argument does not count.]

d) To sum up: The authors need to do more to convince me that the group differences they see are really placebo-specific effects.

2. Robustness

This study is done on a background of consistently negative findings in all five prior studies trying to show non-deceptive placebo effects on objective behavioral or biological measures, as the authors point out in their introduction. The authors argue that these tests were all not sensitive enough to find non-deceptive placebo effect since they are not even sensitive enough to demonstrate deceptive placebo effects (p. 4: "Put simply, if a deceptive placebo induced through verbal suggestion does not reliably impact objective outcomes in these contexts, there is no reason to expect a non-deceptive placebo should either.") I agree. And I applaud the authors for selecting a task (affective picture viewing, combined with EEG) that is likely to have sufficient sensitivity, based on prior successes with the task in detecting deceptive placebo effects (p. 4). Nevertheless, the EEG effects that the authors describe are not internally consistent (relative response dampening by placebo in sustained LPP, response amplification in early LPP, see 1.d), the directionality of placebo effects on EEG markers in the existing deceptive placebo literature is also very inconsistent (p. 10), the direction of the EEG effects in the current study was not predicted, and there is an absence of a to-be-expected three-way interaction (i.e., a picture valence effect) in the sustained EEG data. I also note that the authors have some difficulty in reconciling their divergent findings (p. 8, bottom: "general dampening effect", when referring to the sustained EEG effect, vs. p. 11: increased "willingness to engage with incoming unpleasant Stimuli", when referring to the early EEG effect). To me this suggests that EEG is quite sensitive in picking up something, but as long as there is no consistency within and across studies, this something may as well be a spurious, random effect, i.e., a false positive. This suspicion is, of course, strengthened given the consistent failure in prior work to show non-deceptive placebo effects on objective measures (see above). At the same time, convincingly demonstrating an objective non-deceptive placebo effects would be a big advance for the field and a very important contribution to the literature. Therefore, I think that this paper would massively benefit from a replication study. As a single EEG study, I fear the experiment will never convince enough readers outside a "circle of believers". It will just be another EEG study.

Minor points:

3. p. 7 bottom/p. 8 top: "participants in the non-deceptive placebo group showed a gradual reduction in sustained LPP amplitude over Time". Please make clear right from the start that "time" is time within a trial, not across the experiment.

4. Sometimes, the authors commit the fallacy of reverse inference when they equate a certain EEG phenomenon with a cognitive process, e.g., p. 10 bottom: "we observed a gradual increase in attention allocation to negative images relative to neutral Images" or p. 14: "Participants in the non-deceptive placebo group showed a larger relative increase in attention allocation".

5. p. 15 bottom: "non-deceptive placebos also offer an alternative emotion regulation strategy that is distinct from internally-generated reappraisal strategies, which often require intact cognitive control mechanisms and cognitive effort". How do the authors know that this is a distinct strategy? Can they show that the placebo effect does not require cognitive control and effort? Generally, it seems the authors read a bit too much into their EEG data.

Reviewer #2:

Remarks to the Author:

In this manuscript, authors describe a solid study in a sample of undergraduate students, where non-deceptive placebo and a control condition are compared regarding their impact on cerebral (EEG/ERP) and self-reported emotional engagement. The rationale is clear and solid, stating that previous work on non-deceptive placebos has focused on dependent variables that are not modulated by deceptive placebos either. The present study measures variables that are more related to emotional reactivity or emotion regulation, with the expectation that these will be modulated. The authors find support for the expected result. The picture viewing task appears solid, even though the external validity is somewhat unclear - would a placebo not eventually be used in the context of pain or suffering rather than transient emotional state? The case for this facet of the study could be made more strongly. As-is, the clinical relevance of the findings is somewhat vague, especially as one would expect demand characteristics to prompt participants to produce the subjective report findings, which surprisingly is not discussed in the manuscript. This is an important point which may render one of two dependent variables hard to interpret. One additional major concern arises from the lack of a positive-picture control condition. The authors seem to claim that non-deceptive placebos trigger a process akin to or identical to emotion regulation (incidentally, what would then be the purpose of the placebo? One may just instruct people to feel less then?) of emotional distress. To demonstrate the negative-emotion specificity however, a positive-picture control condition would be needed. In its absence, the data will be difficult to interpret. Further issues are related to the interpretation of the brain data. Is 400-1000 ms still considered an automatic response? Why is contrasted to subsequent conscious processing - is the first second supposed to be unconscious? The statistical analyses are standard and correctly executed. In summary, this is a solid study addressing a very specific problem of interest to researchers with a focus on emotion regulation via the placebo effect. Some concerns arise regarding the limitations of the experimental design, the potential presence of demand-prompted subjective report, and

the interpretation of the brain data.

Reviewer #3:

Remarks to the Author:

This study investigated the effects of a non-deceptive placebo on emotional distress and late electrocortical positivity (late positive potential; LPP). To the best of my knowledge, this is the first EEG experiment on this topic. Therefore, the results are novel and the manuscript is important to placebo researchers. Moreover, the findings have important translational implications (treatment of emotional distress in different health-related contexts).

The authors conducted two experiments with an image viewing task. They administered negative pictures (mutilation, aggression) and neutral pictures (e.g., household items). The pictures were presented with/ without a non-deceptive placebo (nasal spray with a saline solution).

In experiment 1, the group that received the non-deceptive placebo (n = 33 women) reported less emotional distress for the negative pictures than the control group (n = 29 women).

In experiment 2, the group that received the non-deceptive placebo (n = 50) showed a gradual reduction in the amplitude of the LPP for both neutral and negative images relative to the control group (n = 50).

I have the following comments/ questions.

The authors claim that the non-deceptive placebo reduced emotional distress as indexed by self-report and electrocortical responses.

Previous EEG research has consistently shown that late positivity to affective pictures is modulated by the intrinsic motivational significance and the evaluative context of the picture presentation (e.g., Schupp et al., 2000).

Therefore, different interpretations of the observed findings are possible:

1) The authors created two different emotional/ evaluative contexts for the image viewing task. The control group read an article about the neurobiological processes of pain; the placebo group read an article about the placebo effect in different domains. Subsequently, pain-related images were shown (mutilation, aggression). This can be considered negative priming for the control group.

2) The authors combined negative and neutral pictures in order to analyze the LPP.

However, the neutral images do not elicit emotional distress and should not be responsive to the placebo intervention (as shown for the subjective ratings). The neutral condition is the control condition for the placebo effect on negative emotional picture processing.

Please also note: Post-hoc t-tests (comparisons of LPP amplitudes between the placebo and control group for the five time windows of late positivity) are not statistically significant after correction for multiple comparisons.

3) 'Regular/deceptive' placebos have produced heterogeneous electrocortical effects during the processing of negative affective pictures (e.g., decreased P2 amplitude and increased N2 amplitude, see Zhang et al., 2009, increased frontal LPP amplitude (Übel et al., 2015).

Therefore, it would have been important to see a 'regular placebo response' to reference the

response to the non-deceptive placebo (this could be easily introduced as additional condition of the present experiment).

4) To test that the placebo and control group did not differ in key elements, the authors assessed the reading time of the articles, beliefs in the effectiveness of non-deceptive placebos, and personality traits. They did not assess uneasiness about blood and injury (the contents of the negative pictures).

REVIEWER #1

Comment #1: Subjects in the control condition are made to read an article about pain before the experiment. I was wondering if this might induce a general negative emotional state or even negative expectations about the effects of negative or unpleasant emotional material on well-being, distress, or health. In other words: could the control condition effectively be a distress or even nocebo condition? (In which case the differences in ratings and EEG measures between the two groups could be driven by amplifying effects on affective responses in the control group rather than by dampening effects in the placebo condition.)

Response: Experiment 1 assessed participants mood after the manipulation and before the image viewing task. As the revised MS reports, there were no differences in mood as a function of the manipulation (see Supplementary Table 7). We believe this data addresses this concern. The revised MS also reports data indicating that the manipulations were matched on several theoretically relevant dimensions including number of negative words, narrative structure, and valence of information (See Supplementary Methods 1).

Comment #2. In relation to this: I would be more confident that the group differences in rating and EEG are really due to the positive expectations raised by the placebo manipulation if the authors could demonstrate a correlation of rating/EEG effects with individual or group differences in the ratings of subjects' beliefs in the effects of non-deceptive placebos as well as in the effects of the nasal spray that the subjects provided at the end of the instruction phase (see Supplement p. 13 ff.). Supplementary Tables 2 and 3 show that these mostly differ between groups and also exhibit substantial variability within groups, and the numbers of subjects included in the two studies (62 and 100, resp.) are clearly sufficient for performing individual-differences analyses. Hence, these analyses are feasible. Showing a relationship between expectations and placebo effects is also more or less standard in placebo studies.

Response: We do not find a relationship between our belief and expectation questions with self-report/EEG measures. These associations are not commonly observed in non-deceptive placebo studies. For example, of the 26 non-deceptive placebo studies published thus far, only 11 report asking expectation questions (42%) and 2 (8%) report observing a relationship (El Brihi et al., 2019; Meeuwis et al., 2017). This may be a function of the well documented fact that constructs measured across different levels of analysis (e.g., self-report, behavior, physiology, neural) often do not cohere (Dang, King, Inzlicht, 2020; Mauss et al., 2005). It is well established that people lack direct access to internal states (Nisbett & Wilson, 1977) and multiple scholars have noted that implicit/physiological and self-report measures often do not reliably correlate with each other (Evers et al., 2013; Mauss et al., 2005). The revised MS discusses these issues and call for more research on how different levels of analyses cohere (pg. 13).

Comment #3. Another related point: Supplementary Table 3 also shows that subjects in the control group of study 2 perceived experimenters as warmer and more competent. This is a surely unexpected effect of the manipulation that again may have had an impact on EEG in this case. The authors themselves think that this finding suggests "that there may be something about our non-deceptive placebo manipulation (i.e., verbal suggestion instructions during the nasal spray instructions) that decreases perception of warmth and competence. This may have influenced the effectiveness of our manipulation ...". Hence, there is a clear need to disambiguate better the potential sources of the described differential EEG effect, which could be done by some kind of multiple regression including all the different ratings. [Note that the authors believe the reduced perceptions of experimenters' warmth and competence in the placebo group may have "reduced the effects we observed" (Suppl. Table 3). They thereby seem to suggest that this could only have made their EEG-based assay of the placebo effect less sensitive, but not have produced false positive findings. However, their EEG effects go both ways – relatively reduced responding in the placebo group in the sustained LPP (Fig. 2) and relatively enhanced responding in the early LPP (Fig. 3); hence, the argument does not count.

Response: There are no longer differences in perceptions of warmth and competence by Condition in the combined sample (Please see Supplementary Table 8). Moreover, when we included perceptions of experimenter warmth and competence in a model, it did not substantively influence any of the results we report.

Comment #4. Nevertheless, the EEG effects that the authors describe are not internally consistent (relative response dampening by placebo in sustained LPP, response amplification in early LPP, see 1.d), the directionality of placebo effects on EEG markers in the existing deceptive placebo literature is also very inconsistent (p. 10), the direction of the EEG effects in the current study was not predicted, and there is an absence of a to-be-expected three-way interaction (i.e., a picture valence effect) in the sustained EEG data. I also note that the authors have some difficulty in reconciling their divergent findings (p. 8, bottom: "general dampening effect", when referring to the sustained EEG effect, vs. p. 11:

increased "willingness to engage with incoming unpleasant Stimuli", when referring to the early EEG effect). To me this suggests that EEG is quite sensitive in picking up something, but as long as there is no consistency within and across studies, this something may as well be a spurious, random effect, i.e., a false positive. This suspicion is, of course, strengthened given the consistent failure in prior work to show non-deceptive placebo effects on objective measures (see above). At the same time, convincingly demonstrating an objective non-deceptive placebo effects would be a big advance for the field and a very important contribution to the literature. Therefore, I think that this paper would massively benefit from a replication study. As a single EEG study, I fear the experiment will never convince enough readers outside a "circle of believers".

Response: This issue has been addressed, as we replicated our original findings. Please see response to Action Editor Comment #2.

Comment #5. p. 7 bottom/p. 8 top: "participants in the non-deceptive placebo group showed a gradual reduction in sustained LPP amplitude over Time". Please make clear right from the start that "time" is time within a trial, not across the experiment.

Response: We now do this.

Comment #6. Sometimes, the authors commit the fallacy of reverse inference when they equate a certain EEG phenomenon with a cognitive process, e.g., p. 10 bottom: "we observed a gradual increase in attention allocation to negative images relative to neutral images" or p. 14: "Participants in the non-deceptive placebo group showed a larger relative increase in attention allocation".

Response: Thank you for drawing our attention to this. We have corrected this.

Comment #7. p. 15 bottom: "non-deceptive placebos also offer an alternative emotion regulation strategy that is distinct from internally-generated reappraisal strategies, which often require intact cognitive control mechanisms and cognitive effort". How do the authors know that this is a distinct strategy? Can they show that the placebo effect does not require cognitive control and effort? Generally, it seems the authors read a bit too much into their EEG data.

Response: Thank you for drawing our attention to this. We scaled back our interpretation of the EEG data.

REVIEWER # 2

Comment #1: The authors find support for the expected result. The picture viewing task appears solid, even though the external validity is somewhat unclear - would a placebo not eventually be used in the context of pain or suffering rather than transient emotional state? The case for this facet of the study could be made more strongly. As-is, the clinical relevance of the findings is somewhat vague, especially as one would expect demand characteristics to prompt participants to produce the subjective report findings, which surprisingly is not discussed in the manuscript.

Response: Because we were interested in establishing that non-deceptive placebos were real psychobiological effects, we wanted to use a standardized emotion induction task using IAPS pictures, which provide information about online emotional experiences of relevance to a range of physical and psychiatric conditions (Allen, Trinder, Trinder, & Brennan, 1999; de Tommaso et al.,

2009; Shapira et al., 2003). Indeed, there is substantial evidence that IAPS images produce transient emotional states in community and student populations as well as in patients presenting with different physical and psychiatric conditions (Sheppes et al., 2015). As the revised MS notes, an important question for future research to address is how these findings generalize to other kinds of negative affect eliciting stimuli.

Moreover, while we believe that non-deceptive placebos can be used in other clinical contexts such as pain, we believe emotional distress is an important feature of many physical and psychiatric conditions and merit their own intervention (DeSteno, Gross, & Kubzansky, 2013). For example, in terms of physical conditions, emotions play an important role in different types of pain experience (Flaten et al., 2011). In terms of psychiatric conditions, negative emotions and their regulation feature heavily for people who suffer from different types of anxiety disorders and major depressive disorder (Sheppes et al., 2015). We now try to make this clearer to make a stronger case for the importance of studying emotions in the context of non-deceptive placebos.

We also agree with Reviewer #2 that demand characteristics may have produced the subjective report findings in Experiment 1. This is precisely why we ran an experiment that measured neurobiological markers of emotional distress. We believe that the EEG findings from Experiment 2 provides evidence that the differences we find in self-report from Experiment 1 is more than demand characteristic. The revised MS addresses this issue explicitly on page 11.

Comment #2: One additional major concern arises from the lack of a positive-picture control condition. The authors seem to claim that non-deceptive placebos trigger a process akin to or identical to emotion regulation (incidentally, what would then be the purpose of the placebo? One may just instruct people to feel less then?) of emotional distress. To demonstrate the negative-emotion specificity however, a positive-picture control condition would be needed. In its absence, the data will be difficult to interpret.

Response: We have already explained why we did not include a positive-picture-control condition in this paper (see our response to Comment #1 from the Action Editor).

With respect to Reviewer #2 comment concerning “what would then be the purpose of the placebo? One may just instruct people to feel less then?”, we believe there are reasons to believe that placebos, which often require the administration of an external object, are phenomenologically distinct from directly instructing people to feel better. Emotion regulation researchers have speculated that placebos may involve more automatic processes compared to more controlled internal processes like reappraisal (Braunstein, Gross, & Ochsner, 2017). Although we do not test this idea here, there is some previous work suggesting that placebos may require minimal working memory to exert their effect (Buhle et al, 2012) while engaging in cognitive reappraisal tends to be mentally taxing (Schmeichel, Volokhov, & Demaree, 2008). We discuss this work in the revised MS (p. 12)

Comment #3. Further issues are related to the interpretation of the brain data. Is 400-1000 ms still considered an automatic response? Why is it contrasted to subsequent conscious processing - is the first second supposed to be unconscious?

Response: Thank you for this comment. It allowed us to clarify our description of the time windows. We believe that the 400-1000ms and the 1000-6000ms time windows are distinct LPP components based on previous work (Hajcak et al., 2010; Moser et al., 2014, 2017; Shafir 2015, 2018). We believe that the 400-1000ms time window indexes early attentional allocation whereas later time windows such as 1000-6000ms indexes memory and meaning making stages (Moser et al., 2014). The distinction is useful because there are emotion regulation studies showing that some strategies

like distraction tend to affect earlier LPP components (400-1000ms) while other strategies like reappraisal tend to affect later time windows (Qi et al., 2017; Thiruchselvam et al., 2011). We believe that observing non-deceptive placebo effects in these two time windows would be fruitful.

REVIEWER # 3

Comment #1. The authors created two different emotional/ evaluative contexts for the image viewing task. The control group read an article about the neurobiological processes of pain; the placebo group read an article about the placebo effect in different domains. Subsequently, pain-related images were shown (mutilation, aggression). This can be considered negative priming for the control group.

Response: This issue has already been addressed (see our response to Reviewer #1, Comment #2).

Comment #2. The authors combined negative and neutral pictures in order to analyze the LPP. However, the neutral images do not elicit emotional distress and should not be responsive to the placebo intervention (as shown for the subjective ratings). The neutral condition is the control condition for the placebo effect on negative emotional picture processing.

Response: This issue has already been addressed (see Comment #1 from Action Editor).

Comment #3. 'Regular/deceptive' placebos have produced heterogeneous electrocortical effects during the processing of negative affective pictures (e.g., decreased P2 amplitude and increased N2 amplitude, see Zhang et al., 2009, increased frontal LPP amplitude (Übel et al., 2015). Therefore, it would have been important to see a 'regular placebo response' to reference the response to the non-deceptive placebo (this could be easily introduced as additional condition of the present experiment).

Response: While we agree with Reviewer #3's suggestion that a comparison group of deceptive placebo response would be interesting, our goal was to perform a high-powered test of whether non-deceptive placebos can produce objective psychobiological effects. The revised MS highlights the need to calibrate the magnitude of these effects in future work by including a deceptive placebo condition.

Comment #4. To test that the placebo and control group did not differ in key elements, the authors assessed the reading time of the articles, beliefs in the effectiveness of non-deceptive placebos, and personality traits. They did not assess uneasiness about blood and injury (the contents of the negative pictures).

Response: Reviewer #3 is correct in that we did not measure uneasiness about blood and injury unfortunately and therefore cannot covary it out in our model. The fact that we randomly assigned participants to condition should, however, address this concern. We also want to point out that our images were diverse in content with some including blood and injury, but also those consisting of disaster threats, human threats, and human and animal distress. Moreover, we do have other measures that we know individually influence sustained LPP amplitude from these images like trait anxiety, propensity to worry, social desirability responding tendency, and optimism. The two groups did not differ in most of these individual difference variables which we now report in Supplementary Table 8. We did, however, find that people in the non-deceptive placebo group reported marginally more trait anxiety than people in the control group ($p = .07$). When we controlled for it in our overall model, it was a not significant predictor of sustained LPP activity, and it did not significantly impact our main effect and interaction findings. Notably, the fact that the non-deceptive placebo group

showed slightly increased trait anxiety compared to the control group should have worked against reduction in the LPP, but as reported in both samples, we find robust decreases in the sustained LPP for the non-deceptive placebo group.

Reviewers' Comments:

Reviewer #1:

Remarks to the Author:

The authors have done an excellent job in addressing all my concerns. Notably, they have added substantial power to their analysis by significantly enlarging their sample. This not only strengthens confidence into the main findings of the study (on the placebo effect on the late LPP) but it also led to the elimination or reduction of placebo effects on the early LPP, which appeared inconsistent or implausible in the first version of the manuscript. This further underlines the importance of sufficiently powered samples.

I believe that this is now a very solid and at the same time very important paper, and I strongly recommend publication.

One little detail, which I think can be addressed without further involvement of reviewers: In L152, the authors announce to "discuss in more detail below" effects of placebos on reactivity to both neutral and negative emotional stimuli, referring to similar observations in other studies. I could not find such a discussion, though, in the remainder of the paper. I think it would be important to add it, such that the reader has a chance to integrate the current findings into a broader picture.

Reviewer #2:

Remarks to the Author:

The authors should be applauded for replicating their work. Unfortunately, this did not address my concerns, which have in fact been aggravated by their responses. Most importantly, given the sensitivity of the brain potential to regulation strategies, it seems it is now conceptually impossible to know if the brain potential amplitude (susceptible to cognitive elaboration) and self-report (susceptible to elaboration as well) both just reflect demand characteristics. This seems to broadly limit the ability of the data to address the research question. It is a hard question to tackle, and maybe the present measures are not sufficient to come to a clear conclusion. In addition, it may not be appropriate to refer to a time range up to 1000 ms post-stimulus as reflecting "early attention".

REVIEWER # 1

Comment #1: The authors have done an excellent job in addressing all my concerns. Notably, they have added substantial power to their analysis by significantly enlarging their sample. This not only strengthens confidence into the main findings of the study (on the placebo effect on the late LPP) but it also led to the elimination or reduction of placebo effects on the early LPP, which appeared inconsistent or implausible in the first version of the manuscript. This further underline the importance of sufficiently powered samples. I believe that this is now a very solid and at the same time very important paper, and I strongly recommend publication.

Response: We thank Reviewer 1 for their feedback and suggestion of running another sample.

Comment #2: One little detail, which I think can be addressed without further involvement of reviewers: In L152, the authors announce to “discuss in more detail below” effects of placebos on reactivity to both neutral and negative emotional stimuli, referring to similar observations in other studies. I could not find such a discussion, though, in the remainder of the paper. I think it would be important to add it, such that the reader has a chance to integrate the current findings into a broader picture.

Response: Thank you for pointing out this confusion in our wording. We now specifically say, “we elaborate in more detail in the Discussion.” This information is located in the fourth paragraph of the Discussion section. In this paragraph, we point out that some studies have also shown an asymmetry in terms of how the placebo manipulation impacted participants' self-report and physiological responses (Meyer et al., 2015, 2019; Schienle et al., 2020). We speculate that this asymmetry may have to do with the temporal feature of self-report and the sustained LPP. Since the sustained LPP measures online reactions to images, while the self-report is assessed four seconds after picture offset, it may be that by the time participants are asked about the neutral images, any small distress they may have experienced have returned to baseline. We also point out that there is a large body of research suggesting that self-report, behavior, peripheral physiological, and neural measures are not redundant and frequently do not cohere with one another (Mauss et al., 2005; Dan-Glauser & Gross, 2013; Dang et al., 2020), which is why it is important to examine the effects of our intervention across multiple levels of measurements.

Reviewer # 2

Comment #1: The authors should be applauded for replicating their work. Unfortunately, this did not address my concerns, which have in fact been aggravated by their responses. Most importantly, given the sensitivity of the brain potential to regulation strategies, it seems it is now conceptually impossible to know if the brain potential amplitude (susceptible to cognitive elaboration) and self-report (susceptible to elaboration as well) both just reflect demand characteristics. This seems to broadly limit the ability of the data to address the research question. It is a hard question to tackle, and maybe the present measures are not sufficient to come to a clear conclusion.

Response: The LPP is widely used as a biomarker of emotional reactivity in emotion regulation research (Hajcak, MacNamarra, & Olvet, 2010; Proudfit et al., 2014; Thiruchselvam et al., 2011). It is generated by a distributed neural network involving both cortical (e.g., parietal) and subcortical (e.g., amygdala) regions (for a review, see Hajcak & Foti, 2020). Together, the LPP is considered to reflect “stimulus significance and associated activation of motivational circuits” (page 8, Hajcak & Foti, 2020). The effects of emotion regulation strategies on the amplitude of the LPP are therefore seen as having a *direct impact* on the motivational significance of incoming stimuli that is independent of experimenter demand (Moser et al., 2009; Moser et al., 2014; Moser et al., 2017; Shafir & Sheppes, 2018). In fact, there is no research that we are aware of that indicates the LPP is influenced by experimenter demand. It is also unclear how demand would have a significant effect on the LPP, given that it reflects rapid, online brain activity that unfolds over a few seconds in a fast picture viewing task. Like a great deal of prior emotion regulation research (Moser et al., 2009; Moser et al., 2014, Moser et al., 2017; Shafir & Sheppes, 2018; Shafir, Zucker, & Sheppes, 2018; Speed et al., 2017; Thiruchselvam et al., 2011), we conclude that the non-deceptive placebo manipulation had a direct impact on the motivational significance of the incoming stimuli vis-à-vis the decrease of sustained LPP amplitude.

Comment #2: In addition, it may not be appropriate to refer to a time range up to 1000 ms post-stimulus as reflecting "early attention".

Response: We have made changes in the manuscript to reflect that the 400-1000 ms post-stimulus as not reflecting “early attention” but rather broad attentional processes (Moser et al., 2014). This is consistent with a large body of research indicating different mechanisms involved in the 400-100ms and 1000+ time windows of the LPP, with attentional mechanisms more clearly reflected in the former and meaning-making mechanisms more clearly reflected in the latter (Hajcak et al. 2012; Hajcak & Foti, 2020).